# Batch Reverse Osmosis Desalination Modeling under a Time-Dependent Pressure Profile

**DOI:** 10.3390/membranes11030173

**Published:** 2021-02-28

**Authors:** Abdeljalil Chougradi, François Zaviska, Ahmed Abed, Jérôme Harmand, Jamal-Eddine Jellal, Marc Heran

**Affiliations:** 1Civil Engineering Department, Mohammadia School of Engineering, Mohammed V University, Rabat 10090, Morocco; chougradi.abdeljalil@gmail.com (A.C.); abed@ymail.com (A.A.); jellaljamaleddineemi@gmail.com (J.-E.J.); 2Institut Européen des Membranes, UMR 5635, CNRS, ENSCM, University of Montpellier, 34095 Montpellier, France; francois.zaviska@umontpellier.fr; 3LBE, INRAE, Montpellier University, 11100 Narbonne, France; jerome.harmand@inrae.fr

**Keywords:** desalination, reverse osmosis, modeling, optimization, batch system, energy demand

## Abstract

As world demand for clean water increases, reverse osmosis (RO) desalination has emerged as an attractive solution. Continuous RO is the most used desalination technology today. However, a new generation of configurations, working in unsteady-state feed concentration and pressure, have gained more attention recently, including the batch RO process. Our work presents a mathematical modeling for batch RO that offers the possibility of monitoring all variables of the process, including specific energy consumption, as a function of time and the recovery ratio. Validation is achieved by comparison with data from the experimental set-up and an existing model in the literature. Energetic comparison with continuous RO processes confirms that batch RO can be more energy efficient than can continuous RO, especially at a higher recovery ratio. It used, at recovery, 31% less energy for seawater and 19% less energy for brackish water. Modeling also proves that the batch RO process does not have to function under constant flux to deliver good energetic performance. In fact, under a linear pressure profile, batch RO can still deliver better energetic performance than can a continuous configuration. The parameters analysis shows that salinity, pump and energy recovery devices efficiencies are directly linked to the energy demand. While increasing feed volume has a limited effect after a certain volume due to dilution, it also shows, interestingly, a recovery ratio interval in which feed volume does not affect specific energy consumption.

## 1. Introduction

Humanity is facing the challenge of clean water resource depletion as predictions show that half of humanity may live in regions with water stress problem by 2030 [1]. This situation appears worse if other factors are included such as population growth, the evolving economy, water resource pollution and climate change. Consequently, it is critical to find solutions to increase fresh water production and to provide safe drinking water for the world’s growing population while limiting energy requirements.

Seawater desalination has attracted growing attention in the last few decades as an alternative technology for fresh water augmentation. However, seawater desalination inevitably costs significantly more than treatment of any other surface water resource. Indeed, with consideration of the difference in salinity between raw water (seawater) and fresh water (under World Health Organization tap water regulations), desalination induces such a great difference in chemical potential that it inevitably consumes, from a thermodynamics point of view, a high amount of energy to remove dissolved salt.

Nowadays, the most energy-efficient seawater desalination technology is reverse osmosis (RO). This technology has improved considerably in the last five decades and is at present the most developed seawater desalination technology at industrial scale [2]. These improvements are mainly due to the enhancement of membrane performance (with a quite good compromise between permeability and selectivity), pump efficiency and the implementation of energy recovery devices (ERDs), which result in considerable decreases in energy consumption (from approximatively 15 kWh/m^3^ in the early 1970s to less than 2 kWh/m^3^ today). Nevertheless, this consumption can be further reduced by optimizing the pilot design and its associated operating mode. A new trend is to work with the batch system where the recirculation of the rejected brine goes back into the feed tank. This process is named batch and semi-batch RO configurations. Additionally, there is room for improvement from an energetic standpoint considering the size of the feed tank, the profile pressure applied to the ERD and pump efficiencies.

In continuous mode, the feed pressure depends on (1) the desired conversion yield and (2) the salt concentration to guarantee a minimum permeate flow at the end of the spiral wound. Whereas batch RO is, in theory, the only configuration where the required minimum energy is equal to the thermodynamic theoretical minimal specific energy consumption (SEC), by matching/adapting the applied pressure to the increasing osmotic pressure [3]. Thus, by having the possibility to reduce the difference between pump pressure and osmotic pressure, batch RO makes it possible to control and minimize the polarization layer. It has to be noted that, in reality, it is impossible to reach such a limit due to many potential energy losses such as electrical energy conversion into mechanical energy efficiency (pump efficiency) as well as the concentration of polarization (selective mass transfer), pressure loss (friction) and ERD energy loss. However, the minimum practical energy consumption is reduced by nearly 30% when one passes from a continuous RO configuration to a batch configuration (from 1.54 kWh/m^3^ to 1.1 kWh/m^3^) [3].

Indeed, in a conventional continuous configuration (Figure 1, Type A), the pressure is fixed according to the osmotic pressure of the outlet of the last pressure vessel module; this is to satisfy the objective of treatment in terms of water recovery. In the batch configuration, feed water is pumped and contained in a feed tank, which can be pressurized or not depending on the configuration. The feed is, then pumped through a pressurized membrane vessel where the RO filtration occurs. The permeate is recovered, while the retentate is recirculated to the feed tank resulting in an increase of its concentration. This operation, named a pass, is reconducted several times until reaching the desired water recovery. Then the feed tank is emptied (corresponding to the final concentrate) and refilled to start a new cycle. Two configurations can be adapted in batch mode. The first one requires an ERD to recover the pressure and transfer it to the feed stream (Figure 1, Type B), and the second one requires a pressurized feed tank (Figure 1, Type C). This last configuration, with the pressurized tank, seems, at first sight, easier, but it might be very constraining and difficult to set up at a larger scale. The feed pump in Type A delivers constant pressure, while it delivers time variable pressure in the rest of the processes, to keep producing permeate flux as feed osmotic pressure increases with time. ERDs are used to recover energy from the brine in processes A and B. The pressurized tank is schemed as a tank with a piston that retains the brine’s energy, acting as an ERD.

The advantage of working with the batch RO configuration is that the pressure can be modulated and adapted precisely according to the osmotic pressure evolution (Figure 2). Batch RO can operate similar to an N-stage configuration by increasing pressure like a staircase function to overcome increasing osmotic pressure between stages. An alternative pressure profile is to place an osmotic pressure sensor (conductometer) to apply enough pressure that would keep the net driving pressure (NDP; NPD = ΔP − Δπ) constant to maintain a constant flux. Any random pressure profile that is greater than the osmotic pressure would be suitable for the batch RO. Figure 2 was drawn to compare the two different operating modes. The mean permeate flux was fixed at 12 L·m^−2^·h^−1^ for both configurations. The osmotic pressure stays parallel to the pump pressure for batch RO, whereas the osmotic pressure tends to reach the pump pressure for continuous RO. In continuous RO, the permeate flux is also a function of the module position in the pressure vessel (from 26 to 3 LMH), while in continuous RO, it remains constant as the batch RO pressure was set to deliver constant flux. Thus, the recovery ratio depends on the module’s place in the pressure vessel (PV) for continuous configuration, whereas in batch RO it is a function of the process time. What is also important to note is that the salt convective flux (J_S_ = J_W_·C_S_) is different, showing that the scaling risk is not the same.

Another alternative to continuous RO desalination is the semi-batch configuration (Figure 1, Type D). The main difference with the batch process is that the recirculation stream is mixed instantly with the feed stream, instead of being stored in a feed tank. Feed salinity increases with time; thus, the pump pressure also increases to keep a positive permeate flux. While the main focus of our study is the modeling of batch RO desalination, it is worth mentioning that the semi-batch process, also known as closed circuit reverse osmosis (CCRO), is patented and commercialized by Desalitech Company under the name of Reflex CCRO [4]. The company claims a high recovery ratio of up to 98%, energy savings as well as less fouling and scaling. The main findings regarding the performance of CCRO were published in a series of papers exploring all aspects of this technology [5,6,7]. CCRO is now incorporated in different RO software such as ROSA [8], LewaPlus [9] and PROTON [10].

Research on batch configuration is still limited, and large-scale use remains under investigation. Some patents were introduced by Szucz et al. [11], Oklejas [12] and Warsinger et al. [13]. Warsinger et al. [14] modeled the batch configuration and semi-batch configurations and found that they can save up to 64% and 37% of energy, respectively, for brackish water at high water recovery. They explained that the batch configuration exhibits higher energy efficiency than CCRO does because of the high entropy generated in CCRO caused by mixing brine with feed water, which is lessened in the batch process where the concentration difference between the brine and the feed is much lower (both stream concentrations increase). Another advantage of the batch mode is the less fouling propensity due to better control of the effective driving force, which allows to control the polarization concentration phenomena and thus reduces fouling. Warsinger et al. [15] explored the effect of batch configuration on scaling. They concluded that due to the shorter residence time of scalants and the cyclic concentration of the seawater feed, batch RO is more likely to resist inorganic fouling of Gypsum CaSO_4_ and could reach high recoveries greater than 75% while continuous RO is limited to 60% in order to avoid scaling under the same conditions.

Our paper proposes an approach to modeling the batch RO process that is based on the works of Slater et al. [16] and is different than recent models. We opted for this model because it allows the use of time-dependent pressure profiles and detailed description of process variable dynamics, using a forward and direct analytical approach delivering differential equation describing the whole batch RO system concentrations. A Python algorithm was developed to that end. Validation is conducted by comparison to existing models and experimental data. An energetic comparison between batch RO and continuous RO configurations is discussed to highlight energetic performances. The batch RO process is also simulated under a wide set of parameter variations and under different pressure profiles to explore its energetic response.

## 2. Materials and Methods

### 2.1. Experimental Set-Up

A laboratory-scale system developed by Koch (labcell-CF-1 model) was used to conduct the batch RO experiment (Figure 3) in order to properly validate the batch RO model. The selected batch RO configurations used were Type C and B with $\eta_{ERD}$ = 1 (Figure 1).

The set-up was composed of a 500 mL feed tank (pressurized with nitrogen gas), a tangential recirculation pump and a membrane filtration cell. A Filmtec XLE flat sheet RO membrane with a 21.5 cm^2^ surface area was used for this study. The temperature was kept constant at 25 °C by means of a cooling system, and the pressure was imposed manually by controlling the opening valve of the nitrogen pressurized bottle. Permeate was retrieved in a storage recipient while brine was put back into the feed tank by a recirculation pump. Permeate volume was monitored during the process by measuring water weight over time.

Water permeability was identified by measuring the pure water flowrate at different pressures. The value found was 11.08 L/m^2^/h/bar. Salt permeability was determined by measuring permeate salinity with a conductometer for different pressures. The mean value found was 4.12 × 10^−7^ m/s. Two experiments were conducted with different initial feed concentrations and different feed volumes. Pressure was regulated manually; a staircase pressure function of 10 bar + 2 bar/15 min was selected. The experiments were conducted for an average time of 105 min, and permeate average concentration and its weight were measured every 5 min.

### 2.2. Process Modeling

Batch RO with a non-pressurized feed tank was considered for this modeling study (Type B), which can also be suitable for Type C by fixing $\eta_{ERD}$ = 1. Figure 4 represents the scheme of a batch RO configuration. The raw water is stored in a feed tank (with volume *V_f_* [*t*]) and goes through the RO membrane. The produced water is stored in a permeate tank (with volume *V_p_* [*t*]) whereas the concentrate is recirculated to the feed tank. The feed tank volume decreases with time as the feed goes through an RO membrane (permeate). Feed tank concentration *C_f_* (*t*) increases because of the retentate recirculation. The permeate tank receives produced water at each pass at concentration *C_p_* (*t*) where the average concentration is denoted *C_pav_*(*t*) and the permeate volume is denoted *V_p_*(*t*). The pump is of variable pressure and can be adjusted as desired to overcome feed osmotic pressure increases, with a constant flowrate *Q_f_* (*t*). The permeate flowrate is denoted *Q_p_*(*t*). The retentate flowrate is given by *Q_r_*(*t*) = *Q_f_* (*t*) − *Q_p_*(*t*). The concentrate energy is retrieved via an ERD and channeled to the feed stream. This configuration is equivalent to batch RO with a pressurized feed tank if the ERD efficiency is ideal.

The RO module is characterized by its surface area S, water permeability $A_{w}$ and salt permeability $B_{s}$. A cycle is composed of several passes, and it ends when a condition of one of the variables is reached, for example, the feed osmotic pressure or recovery ratio. The feed tank is then emptied and refilled to start another cycle. All concentrations, volumes, pressures and fluxes depend on time due to the transient nature of the batch process. With consideration that the recovery ratio per pass is very low compared to the recovery ratio per cycle, the spatial osmotic variation inside the membrane was neglected. Moreover, the batch RO process is usually constituted of only one or two elements per pressure vessel [18,19], thereby limiting the spatial osmotic variation. The feed osmotic pressure is time dependent due to the recirculation of concentrate which will also impact the inlet tank osmotic pressure. The following initial conditions are adopted:$$V_{f}\left(t=0\right)=V_{f0},\;C_{f}\left(t=0\right)=C_{f0},\;C_{p}\left(t=0\right)=C_{pav}\left(t=0\right)=0.$$

#### 2.2.1. Fluxes Models

Van’t Hoff’s law is used to express osmotic pressure π as a function of feed concentration (Equation (1a,b)). The produced water is estimated through the solvent flux $J_{w}\;$(L/h/m^2^), whereas its quality is linked to the salt transport/transfer solute flux $J_{s}$ (kg/h/m^2^) as shown in Equations (2) and (3). Equation (4) gives the relation between the water production and its quality. The pressure drop along the spiral wound is noted as ΔL (Equation (5)).
(1a)π(t)=φn(t)RT=ψC(t),
(1b)$$\Delta \pi \left(t\right)=\;\psi (C_{f}\left(t\right)-C_{p}\left(t\right)),$$
(2)$$J_{w}\left(t\right)\;=\;A_{w}\;\left(\Delta P\left(t\right)-\Delta \pi \left(t\right)\right),$$
(3)$$J_{s}\left(t\right)\;=\;B_{s}\;(C_{f}\left(t\right)-C_{p}\left(t\right)),$$
(4)$$J_{s}\left(t\right)=J_{w}\left(t\right)C_{p}\left(t\right)$$
and
(5)$$\Delta P\left(t\right)=\;\Delta P_{pump}\left(t\right)-\Delta L/2$$
where ΔP(t) represents both the booster pump pressure and main pump pressure.

#### 2.2.2. Mass Balances

The mass balance made on the product tank gives
$$Q_{p}\left(t\right)C_{p}\left(t\right)=\frac{d\left(V_{p}\left(t\right)C_{pav}\left(t\right)\right)}{dt}=\frac{dV_{p}\left(t\right)}{dt}C_{pav}\left(t\right)+\frac{dC_{pav}\left(t\right)}{dt}V_{p}\left(t\right),$$
while it is known that $Q_{p}\left(t\right)=\frac{dV_{p}\left(t\right)}{dt}$, the mass balance becomes:(6)$$\frac{dC_{pav}\left(t\right)}{dt}=\frac{Q_{p}\left(t\right)}{V_{p}\left(t\right)}\left(C_{p}\left(t\right)-C_{pav}\left(t\right)\right).$$

The mass balance made on the membrane module gives
(7)$$Q_{p}\left(t\right)C_{p}\left(t\right)=Q_{f}\left(t\right)C_{f}\left(t\right)-Q_{r}\left(t\right)C_{r}\left(t\right).$$

The mass balance made on the feed tank gives
(8)$$Q_{r}\left(t\right)C_{r}\left(t\right)-\;Q_{f}\left(t\right)C_{f}\left(t\right)=\;\frac{d\left(V_{f}\left(t\right)C_{f}\left(t\right)\right)}{dt}$$

As the feed tank is supposed to be a perfectly mixed tank reactor, with a combination of Equations (7) and (8) and with the knowledge that $Q_{p}\left(t\right)=-\frac{dV_{f}\left(t\right)}{dt}$, the variation of the feed concentration is given by
(9)$$\frac{dC_{f}\left(t\right)}{dt}=\frac{Q_{p}\left(t\right)}{V_{f}\left(t\right)}\left(C_{f}\left(t\right)-C_{p}\left(t\right)\right)$$

In this constant volume system (Batch reactor), the global mass balance yields
$$V_{f0}C_{f0}=(V_{f0}-V_{p}\left(t\right))\;C_{f}\left(t\right)+V_{p}\left(t\right)C_{pav}\left(t\right),$$
$$C_{f0}=(1-\frac{V_{p}\left(t\right)}{V_{f0}})\;C_{f}\left(t\right)+\frac{V_{p}\left(t\right)}{V_{f0}}C_{pav}\left(t\right)$$
and
(10)$$X=\frac{V_{p}\left(t\right)}{V_{f0}}=\frac{C_{f}\left(t\right)-C_{f0}}{C_{f}\left(t\right)-C_{pav}\left(t\right)}.$$

In the same way, the fact that the total water volume is constant gives
(11)$$V_{p}\left(t\right)+V_{f}\left(t\right)=V_{f0.}$$

Equations (6) and (9) together form a coupled nonlinear differential system of Equation (12):(12)$$\{\begin{array}{c} \frac{dC_{pav}\left(t\right)}{dt}=\frac{Q_{p}\left(t\right)}{V_{p}\left(t\right)}\left(C_{p}\left(t\right)-C_{pav}\left(t\right)\right)\\ \frac{dC_{f}\left(t\right)}{dt}=\frac{Q_{p}\left(t\right)}{V_{f}\left(t\right)}\left(C_{f}\left(t\right)-C_{p}\left(t\right)\right)\; \end{array}$$

Solving Equation (12) would allow one to find $C_{f}\;$and $C_{pav}$, from which the rest of the variables can be deduced. To do so, $C_{p}$, $Q_{p}$, $V_{p}$ and $V_{f}\;$should be written as a function of $C_{f}$ and $C_{pav}.$

Combining Equations (1)–(5) yields
$$B_{s}\left(C_{f}\left(t\right)-C_{p}\left(t\right)\right)=C_{p}\left(t\right)A_{w}\left[\Delta P\left(t\right)-\psi \left(C_{f}\left(t\right)-C_{p}\left(t\right)\right)\right],$$
which can be rewritten as
$$\frac{A_{w}\psi }{B_{s}}C_{p}{\left(t\right)}^{2}+\left(\frac{A_{w}\left(\Delta P\left(t\right)-\psi C_{f}\left(t\right)\right)}{B_{s}}+1\right)\;C_{p}\left(t\right)-C_{f}\left(t\right)\;=\;0.$$

Solving the second order equation and keeping only a positive solution yields $C_{p}\;$as a function of $C_{f}$ at any given time:$$C_{p}\left(t\right)=\frac{\sqrt{{\left(\frac{A_{w}\left(\Delta P\left(t\right)-\psi C_{f}\left(t\right)\right)}{B_{s}}+1\right)}^{2}+\frac{4A_{w}\psi C_{f}\left(t\right)}{B_{s}}}-\left(\frac{A_{w}\left(\Delta P\left(t\right)-\psi C_{f}\left(t\right)\right)}{B_{s}}+1\right)}{\frac{2A_{w}\psi }{B_{s}}}$$

The following constants are introduced to simplify the expressions:$$\alpha_{1}=A_{w}S,\alpha_{2}=A_{w}S\psi ,\alpha_{3}=\frac{A_{w}}{B_{s}},\alpha_{4}=\frac{A_{w}\psi }{B_{s}},\alpha_{5}=V_{f0}.$$

Further, $C_{p}$ can then be rewritten as
(13)$$C_{p}\left(t\right)=\frac{\sqrt{{\left(\alpha_{3}\Delta P\left(t\right)-\;\alpha_{4}C_{f}\left(t\right)+1\right)}^{2}+4\alpha_{4}C_{f}\left(t\right)}-\;\left(\alpha_{3}\Delta P\left(t\right)-\;\alpha_{4}C_{f}\left(t\right)+1\right)\;}{2\alpha_{4}}.$$

Additionally, $Q_{p}$ can be deduced from Equation (2):(14)$$Q_{p}\left(t\right)=J_{w}\left(t\right)S=A_{w}S\left(\Delta P\left(t\right)-\psi \left(C_{f}\left(t\right)-C_{p}\left(t\right)\right)\right)=\alpha_{1}\Delta P\left(t\right)-\alpha_{2}(C_{f}\left(t\right)-C_{p}\left(t\right)).$$

Equations (10) and (11) allow $V_{p}$ and $V_{f}$ to be calculated as a function of $C_{f}$ and $C_{pav}$:(15)$$V_{p}\left(t\right)=V_{f0}\left(\frac{C_{f}\left(t\right)-C_{f0}}{C_{f}\left(t\right)-C_{pav}\left(t\right)}\right)=\alpha_{5}\left(\frac{C_{f}\left(t\right)-C_{f0}}{C_{f}\left(t\right)-C_{pav}\left(t\right)}\right)$$
and
(16)$$V_{f}\left(t\right)=V_{f0}-V_{p}\left(t\right)=\alpha_{5}\left(1-\frac{C_{f}\left(t\right)-C_{f0}}{C_{f}\left(t\right)-C_{pav}\left(t\right)}\right)$$

The coupled differential Equation (12) becomes
$$\{\begin{array}{c} \frac{dC_{pav}\left(t\right)}{dt}=\frac{\left(\alpha_{1}\Delta P\left(t\right)-\;\alpha_{2}(C_{f}\left(t\right)-C_{p}\left(t\right))\right)}{\alpha_{5}\left(\frac{C_{f}\left(t\right)-C_{f0}}{C_{f}\left(t\right)-C_{pav}\left(t\right)}\right)}\left(C_{p}\left(t\right)-C_{pav}\left(t\right)\right)\;\\ \frac{dC_{f}\left(t\right)}{dt}=\frac{\left(\alpha_{1}\Delta P\left(t\right)-\;\alpha_{2}(C_{f}\left(t\right)-C_{p}\left(t\right))\right)}{\alpha_{5}\left(1-\frac{C_{f}\left(t\right)-C_{f0}}{C_{f}\left(t\right)-C_{pav}\left(t\right)}\right)}\left(C_{f}\left(t\right)-C_{p}\left(t\right)\right) \end{array}.$$

Solving the coupled differential equations allows one to compute all variables of the process since they are all written as a function of $C_{f},\;C_{p}$, $C_{pav},\;\Delta P$ and $\alpha_{i}$, making it possible to precisely monitor the whole process.

The input parameters are divided into design parameters (initial tank volume $V_{f0}$ and raw water quality $C_{f0}$), membrane properties (surface S and water and salt permeabilities $A_{w}\;$and $\;B_{s}$) and the operating parameter (membrane pressure over time ΔP(t)).

A Python algorithm was developed to solve Equation (12) and to compute the variables over time using the Runge-Kutta fourth order method. This method was chosen because it does not require higher order derivatives of functions. Moreover, it has a total truncation error on the order of O(h4) (where h is the step size). It may be checked that there is no improvement in the accuracy of the computed trajectories (and thus of the values *C_f_* and *C_pav_*) in using smaller time. Additionally, in this case, the computed values were inside the stability region of the RK4 method. The employed numerical solver offered both stability and convergence, therefore, it was deemed suitable to solve the studied model.

By entering the initial values, membrane characteristics and the pressure function, Equation (12) allows for the computation of $C_{f}$ and $C_{pav}$ for the next pass, which allows the use of Equation (13), (1), (14) and (10) to compute, respectively, the instantaneous permeate concentration, osmotic pressure, permeate flowrate and recovery ratio. The same steps are reconducted until a condition on the permeate average concentration or recovery ratio is reached.

#### 2.2.3. Concentration Polarization

Concentration polarization (CP) is a complex phenomenon in which ions accumulate onto the membrane surface due to the convective flux $J_{w}$. This phenomenon occurs at the interface of the feed solution and the membrane, and it leads to an increase of the local salt concentration (i.e., osmotic pressure), reducing the effective driving force and thus resulting in flux decline and more energy consumption [20]. This increase in the local osmotic pressure generates a CP layer. The film model, used for modeling, supposes a one-dimensional flow, and a concentrating polarization layer only based on$\;C_{f}$ as the conversion yield for one pass is weak in the batch RO process. Thus, with consideration of a boundary layer with a thickness of δ, interfacial membrane concentration $C_{f}$ is the solution of Equation (17),
(17)$$\frac{C_{m}-C_{p}}{C_{f}-C_{p}}=\exp\left(\frac{\;J_{v}}{k}\right);\;J_{v}=\;A_{w}\;\left(\Delta P\left(t\right)-\psi \left(C_{m}-C_{p}\right)\right);CPF=\frac{C_{m}}{C_{f}},$$
where *CPF* is the concentration polarization factor, $C_{m}$ is the solute concentrations at the membrane surface, ψ allows for the osmotic pressure calculation from $C_{m}$ and *k* represents the mass transfer coefficient.

It is common to estimate the mass transfer coefficient using the Sherwood correlation, which illustrates its dependence on the Reynolds number, Schmidt number and Sherwood number. The literature offers a variety of correlations depending on the hydrodynamics regime and the geometric channel design. For the turbulent flow, a widely used Sherwood correlation [21] is shown in Equation (18):(18)$$Sh=\frac{k\;d_{h}}{D}=0.023Re^{0.8}Sc^{0.33}=0.023{\left(\frac{\;ud_{h}}{v}\right)}^{0.8}{\left(\frac{v}{D}\right)}^{0.33}.$$

However, variables, which are usually constant in continuous mode, depend now on time as seen in model equations. Thus, since the feed concentration, permeate concentration and permeate flux depend on time, the concentration on the membrane interface $C_{m}$ will depend on time as well. For each iteration, Equation (19) will be solved after Equation (12) is solved. This means that CPF is a time variant:(19)$$\frac{C_{m}\left(t\right)-C_{p}\left(t\right)}{C_{f\left(t\right)}-C_{p}\left(t\right)}=\exp\left(\frac{\;J_{v}\left(t\right)}{k}\right)CPF\left(t\right)=\frac{C_{m}\left(t\right)}{C_{f}\left(t\right)}.$$

#### 2.2.4. Model Including Concentration Polarization

To include the CP phenomenon in the algorithm, the coupled equations in Equation (12) are changed as the effect of CP on the produced permeate volume $V_{p\_CP}$ and permeate concentration$\;C_{p\_CP}$ is included. The algorithm would also change to compute membrane concentration at each iteration and to take into account the effect of CP on osmotic pressure, permeate flux, permeate concentration, permeate produced volume and recovery ratio. The equations become
$$\begin{array}{c} \frac{dC_{pav\_CP}\left(t\right)}{dt}=\frac{Q_{p\_CP}\left(t\right)}{V_{p\_CP}\left(t\right)}\left(C_{p\_CP}\left(t\right)-C_{pav\_CP}\left(t\right)\right)\\ \frac{dC_{f\_CP}\left(t\right)}{dt}=\frac{Q_{p\_CP}\left(t\right)}{\left(V_{f0}-V_{p\_CP}\left(t\right)\right)}\left(C_{f\_CP}\left(t\right)-C_{p\_CP}\left(t\right)\right)\; \end{array}.$$

At the beginning of each iteration, after $C_{f}$ is computed, $C_{p}$ is deduced using Equation (13), and then $C_{m}$ is computed using Equation (17). The mass transfer coefficient and permeate flux are computed using Equations (18) and (2).

The new permeate concentration which takes into account CP is computed using Equation (13), replacing $C_{f}$ with $C_{m}$:$$C_{p\_CP}\left(t\right)=\frac{\sqrt{{\left(\alpha_{3}\Delta P\left(t\right)-\;\alpha_{4}C_{m}\left(t\right)+1\right)}^{2}+4\alpha_{4}C_{m}\left(t\right)}-\;\left(\alpha_{3}\Delta P\left(t\right)-\;\alpha_{4}C_{m}\left(t\right)+1\right)\;}{2\alpha_{4}}.$$

The new osmotic pressure, flowrate, permeate produced volume and recovery ratio become
$$\Delta \pi_{CP}\left(t\right)\;=\;\psi \left(C_{m}\left(t\right)-C_{p\_CP}\left(t\right)\right),$$
$$Q_{p\_CP}\left(t\right)=J_{w\_CP}\left(t\right)S\;=\;A_{w}S\left(\Delta P\left(t\right)\;-\psi \left(C_{m}\left(t\right)-C_{p\_CP}\left(t\right)\right)\right)\;=\alpha_{1}\Delta P\left(t\right)-\alpha_{2}(C_{m}\left(t\right)-C_{p\_CP}\left(t\right))$$
$$V_{p\_CP}\left(t\right)=\;\int_{0}^{t}Q_{p\_CP}\left(x\right)dx,$$
$$V_{r\_CP}\left(t\right)=\;\int_{0}^{t}\left(Q-\;Q_{p\_CP}\left(x\right)\right)dx,$$
and
$$X_{CP}\left(t\right)=\;\frac{V_{p\_CP}\left(t\right)}{V_{f0}}.$$

The modified equations take into account CP in each iteration for further precise results. The process is reconducted at each iteration as depicted in Figure 5. The inputs are the initial water concentration *C_f_*_0_, the initial water volume in the tank *C_f_*_0_, the pressure function Δ*P*(*t*), the membrane surface area *S* and the water and salt permeabilities *A_W_* and *B_S_*. The step of the returning process variables delivers the value of all variables at any moment. The stop condition depends on the fixed objectives (the recovery ratio or feed concentration in the tank).

#### 2.2.5. Specific Energy Consumption

The SEC is the energy required to produce one cubic meter of permeate. Equation (19) computes the SEC at any moment while the process is running by calculating the sum of the energy used to produce permeate volume $V_{p\_CP}\;$under pressure$\;\Delta P_{pump}$, minus the energy that is not recovered in the concentrate by the ERD when recycling the brine volume (17). The whole is then divided by the pump efficiency and reduced to the produced permeate volume. The ERD energy efficiency is defined by $\eta_{ERD}$.

The developed Python algorithm can compute the SEC at any time in the process. The pressure function, initial values and membrane characteristics are implemented at the beginning of the algorithm. Coupled differential Equation (12) is solved using the Runge-Kutta fourth order to compute $C_{f}$ and $C_{pav}$ of the same iteration, and $C_{p}$ is deduced from Equation (13). Further, $C_{m}$ is deduced using CP Equation (17). Additionally, $C_{m}$ and the newly computed $C_{p\_CP}\;$allow using Equations (14)–(19) to compute process variables while taking into account the CP phenomenon. The same iteration is reconducted until a stopping condition is reached:(20a)$$SEC\left(t\right)=\frac{1}{\eta_{pump}V_{p_{CP}}\left(t\right)}\left[\int_{0}^{V_{p_{CP}}\left(t\right)}\Delta P_{pump}\left(t\right)dV+\int_{0}^{V_{r_{CP}}\left(t\right)}\Delta P_{pump}\left(t\right)dV-\eta_{ERD}\int_{0}^{V_{r_{CP}}\left(t\right)}\left(\Delta P_{pump}\left(t\right)-\frac{\Delta L}{2}\right)dV\right]$$
$$dV_{p\_CP}\left(t\right)=\;Q_{p\_CP}\left(t\right)dt=(\alpha_{1}\Delta P\left(t\right)-\;\alpha_{2}(C_{m}\left(t\right)-C_{p\_CP}\left(t\right)))\;dt$$
$$dV_{r\_CP}\left(t\right)=\;(Q-\;Q_{p\_CP}\left(t\right))\;dt=(Q-\;(\alpha_{1}\Delta P\left(t\right)-\;\alpha_{2}(C_{m}\left(t\right)-C_{p\_CP}\left(t\right))))\;dt$$
(20b)$$SEC\left(t\right)=\frac{1}{\eta_{pump}V_{p\_CP}\left(t\right)}\left[\int_{0}^{t}\Delta P_{pump}\left(x\right)Q_{p\_CP}\left(x\right)dx\;\;+\;\int_{0}^{t}\Delta P_{pump}\left(x\right)\left(Q-Q_{p\_CP}\left(x\right)\right)dx-\eta_{ERD}\int_{0}^{t}\left(\Delta P_{pump}\left(x\right)-\Delta L/2\right)\left(Q-Q_{p\_CP}\left(x\right)\right)dx\right]$$

## 3. Results

### 3.1. Model Validation: Comparison with Experimental Results

Experimental data from the experimental set-up were used to verify the model. The first case study was done a simple aqueous salt (NaCl) solution of 3 g/L whereas the second case was performed with 6 g/L. The initial feed tank was set at 0.4 L, and a staircase pressure function of 10 bar + 2 bar/15 min was selected.

Figure 6a,d compare cumulative permeate variation results over time between the experiment and model simulation for the first and second case (the different initial feed concentrations are, respectively, 3 g/L and 6 g/L). The results allowed the determination of the mass transfer coefficient (*k*). A similar evolution of permeate production at the start of the process was observed, and both plots showed a good agreement between the model and experiment.

Figure 6b,c,e and f show for both cases the osmotic pressure modeling as well as the permeate flux over time and the feed pressure. Pressure plots show how the osmotic pressure tends to reach the pump pressure at the end of the experiment where the feed volume is low leading to high concentration variation. The discontinuities of flux are due to the sudden change in pump pressure since it is a staircase function, which causes the NDP to change suddenly as well. The permeate flux calculation confirms that even with a staircase function, the response is not a monotonic function. In fact, the staircase function leads to a wide permeate flux variation: for the case 1, permeate flux starts at 87 LMH, goes up to 140 LMH and decreases to 10 LMH at the end of process. Similarly, for the case 2, it starts at 60 LMH, goes up to 110 LMH and decreases to 20 LMH. Those variations are not in the favor of the membrane lifetime.

The recovery ratio rate over time was not the same for both cases; it was higher for the case with a lower initial feed salinity. This was explained by the salinity increase rate in the feed tank. If the initial salinity is higher, then osmotic pressure would increase rapidly, and under the same pressure profile and feed tank volume, the production is lower as the NDP is lower.

### 3.2. Validation by Comparison with Wei et al.

The model developed in the present study was compared to the results of a previous model [14,18], validated by the experimental batch RO set-up with a pressurized feed tank. The same conditions displayed in Table 1 were considered except for a slight difference in the CPF, which was hard to adjust as the *k* value was not given.

The energetic performances of both models were compared for fifteen cases, each with different permeate fluxes, initial feed concentrations and recovery ratios. SEC was computed according to Equation (19). The results are depicted in Table 2. The model comparison showed that the SEC estimations are in line with the reference model estimation [14,18] with a maximum error of 3.2% (Equation (21)):Error (%) = 100 × (Wei model − present model)/Wei model(21)

The present model indeed tended to slightly overestimate SEC in most cases compared to the reference model. For lower recovery ratios, there was less difference, while the error increased slightly when the recovery ratio increased. CPF was time dependent, and its mean value in time over the range of 0% to 60% was calculated. Its impact depended on permeate flux: The higher the permeate flux, the higher the value of CPF was.

This difference in energetic performance is mainly due to the difference in the models’ equations and hypotheses. While the reference model [14,18] uses a finite elements method for calculations, we used an analytical approach with coupled differential equations in order to solve the equation step by step. Additionally, this approach considers CPF to be time dependent, and only the mean value is taken into consideration for comparison. Differences in CP values impact the results of the SEC, as Equation (19) uses wall concentration values at each step to compute SEC. Nevertheless, the whole approach tends to deliver energetic performance in good agreement with the reference model.

### 3.3. Batch RO vs. Continuous RO: Energetic Comparison

In this part, a comparison between batch and continuous configurations from an energetic stand point is explored for seawater and brackish water. The comparison was based on the present validated model for batch RO and on the SEC expression for continuous RO [22]. The used parameters are depicted in Table 3. The feed pressure for the batch RO was calculated to work with a constant permeate flux (15 LMH for seawater and 25 LMH for brackish water).

The SEC of batch RO and continuous RO, both using an ERD, over a wide range of recovery ratios are displayed in Figure 7. If batch RO was more energy efficient than continuous RO was at all recovery ratios for seawater, for brackish water the batch RO was more energy efficient only when the recovery ratios is higher than 50% which is always the case for brackish water. The energetic trends are in accordance with the work of Wei et al. [18], Werber et al. [3] and Warsinger et al. [14]. For seawater, at recovery ratios of 40%, 50% and 60%, batch RO used respectively 17%, 23% and 31% less energy than did continuous RO. For brackish water, at recovery ratios of 60%, 70% and 80%, batch RO used respectively 9%, 19% and 34% less energy than did continuous RO.

### 3.4. Batch RO and Pressure Profiles for Seawater

As stated in the introduction, batch RO can operate using any type of pressure profile as long as it remains higher than osmotic pressure during the process. For this part, batch RO pressure profiles were investigated. A linear pressure profile, pressure (*t*) = 35 + 30 *t* (where *t* is the operating time in hours), was compared to batch RO under a pressure profile that delivers a constant flux at 15 LMH and a constant pressure for continuous RO. The membrane parameters are the same as in Table 3. Figure 8a shows pressure profile plots against time, while Figure 8b displays energetic responses over recovery ratios.

Both batch RO pressure profiles used considerably less energy than continuous mode at all recoveries especially at high recovery values. Batch RO under the linear pressure profile performed well, especially at recoveries lower than 50% where its SEC was closed to batch RO under a constant flux pressure profile.

As shown in this example, batch RO energy consumption can increase and perhaps would also decrease with a different pressure profile. This proves that the batch RO still needs some improvement to minimize its SEC.

### 3.5. Impact of Feed Salinity and Feed Volume

Initial feed salinity and feed volume variations that impact batch RO energetic behavior are analyzed in this part. Nine feed tanks (from 0.25 to 20 m^3^: 0.48 L/m^2^ of membrane to 38.6 L/m^2^) and four cases with different feed salinities (32.5 g/L, 35 g/L, 37.5 g/L and 40 g/L) were studied (see Figure 9). The membrane permeability was kept to 3 LMH/bar, and the feed pressure was adjusted to deliver a permeate flux of 15 LMH. The ERD efficiency and the pump efficiency were respectively 97% and 85%. The feed tank’s draining and filling were not considered.

The batch RO process was also run under different feed volumes (Figure 9a). Surprisingly, feed volume seemed to impact batch RO energetics at recovery ratios lower than 50% and higher than 70%, and the feed volume curiously had a zone of zero impact between 55% and 70% recovery ratios. The impact was considerable at recovery ratios below 30% and between feed volumes of 0.25 and 2 m^3^. After a recovery ratio of 70%, the order of SEC plots was interestingly inversed.

The energetic response was directly linked to salinity variation. The higher the feed salinity is, the higher the SEC is. Additionally, for all recovery ratios, the difference between SECs was the same for all recovery ratios. It also seems that increasing salinity with a constant of 2.5 g/L causes an SEC increase with an almost constant value 0.15 kWh/m^3^, which increases with the recovery ratio. This behavior is probably unique to the pressure profile which delivers constant pressure, and it could be that, under a different pressure profile, the variation would have a different energy increase pattern.

Feed volume impact on batch RO is, however, unexpected. While we can explain why lower initial volumes cause SEC to increase (because it causes osmotic pressure to increase quickly while, at higher volumes, dilution slows its increase), it is not clear to us why in a specific interval energetic performance is the same for all volumes and why energy plots are inversed outside this interval.

## 4. Discussion

The energetic gain is mainly due to type of pump pressure profile. The variation of pressure in the batch RO mode, by adjusting the feed pressure according to the osmotic pressure increase in order to keep a constant NDP and also to produce the fixed flux, is the key advantage. In contrast, in the continuous configuration, the pump pressure remains high and constant for continuous RO, which causes SEC to be higher due to the high CPF value in the first module (Figure 10). These promising energetic savings are the main reason behind the growing interest in the batch RO configuration.

The concentration polarization factor represents a loss of driven forces. This loss of energy is more or less constant for batch experiment where a slight decline can be observed on high recovery ratio as the concentration in the permeate increases (Figure 10). The extra power needed to overcome the CPF is $\psi \;CPF\;J_{W}$ for the batch mode whereas its value is $\int \psi \;CPF\left(x\right)J_{w}\left(x\right)$. As the permeat production for the continuous mode is mainly performed on the first pressure vessel modules, this mode is more impacted by the CPF.

## 5. Conclusions

A mathematical model was proposed to simulate the innovative Batch configuration RO. This model has been validated in accordance with experimental data but also with Wei batch model [18] results. Thus, it made it possible to explore the impact of operating and design parameters. As expected, energetic comparison between batch RO and continuous RO proved that the batch configuration is energy efficient especially at higher conversion rate reducing energy consumption of desalination and thus, its environmental impact and costs.

The better energetic gain was due mainly to the variable pressure, which was precisely adjusted to deliver needed energy to produce desired permeate flux and to control the CPF. The control of the CPF value is possible in Batch mode whereas the continuous mode is significantly affected by the CPF on the first pressure vessel modules. Moreover, the control of the CPF value allows the control of the water quality. Then, several parameters impacted batch RO were investigated. Salinity increase caused the SEC to increase in a steady pattern for a rate of 0.15 kwh/m^3^ for every 2.5 g/L increase. On the other side, the increase of the feed volume has a positive impact but beyond 3.86 L/m^2^_Membrane_ the energetic gain is negligible. Surprisingly, there was a recovery ratio interval (55–70%) where feed volume didn’t impact SEC at all.

The next decade is likely to witness a rise in the development and investigation of batch processes, thanks to their promising energy efficiency. Further research prospects can include investigation of Batch RO fouling and CP behaviors, impact of stopping time between batch cycles, establishment of detailed cost estimation, and optimization of process variables.

## Figures and Tables

**Figure 1 membranes-11-00173-f001:**
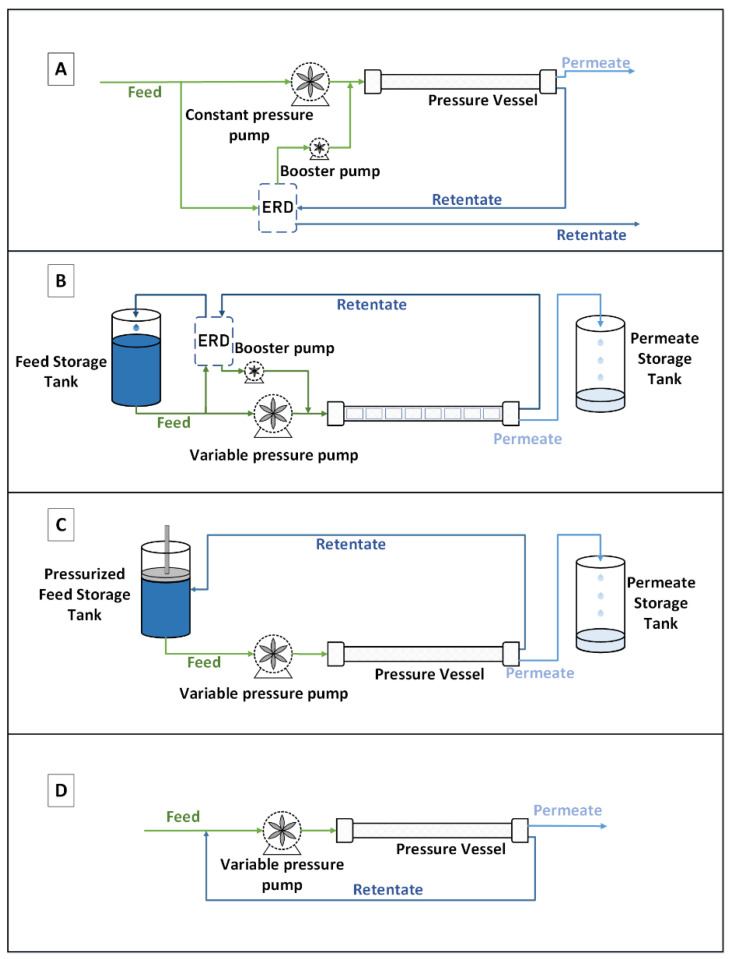
Schemes of four reverse osmosis (RO) desalination processes: Type A, one-stage continuous RO; Type B, batch RO with ERD and non-pressurized feed tank; Type C, batch RO with pressurized feed tank; Type D, semi-batch RO.

**Figure 2 membranes-11-00173-f002:**
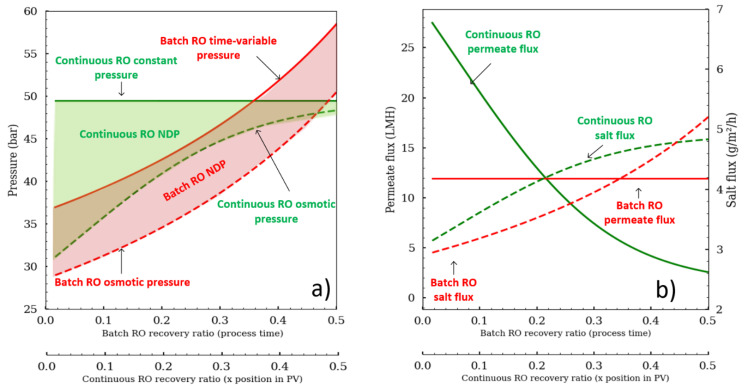
Batch RO and continuous RO (**a**) feed pump pressure and osmotic pressure and (**b**) permeate flux and convective salt flux. (Initial conditions: Salinity = 35 g/L; mean permeate flux = 12 LMH.).

**Figure 3 membranes-11-00173-f003:**
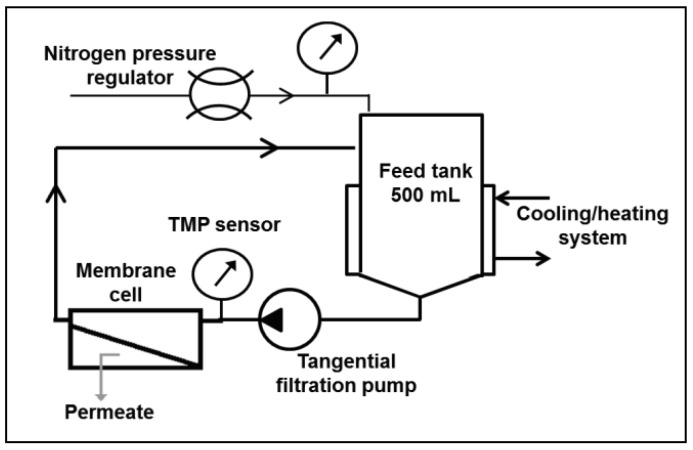
Experimental set-up used to validate the batch RO model [17].

**Figure 4 membranes-11-00173-f004:**
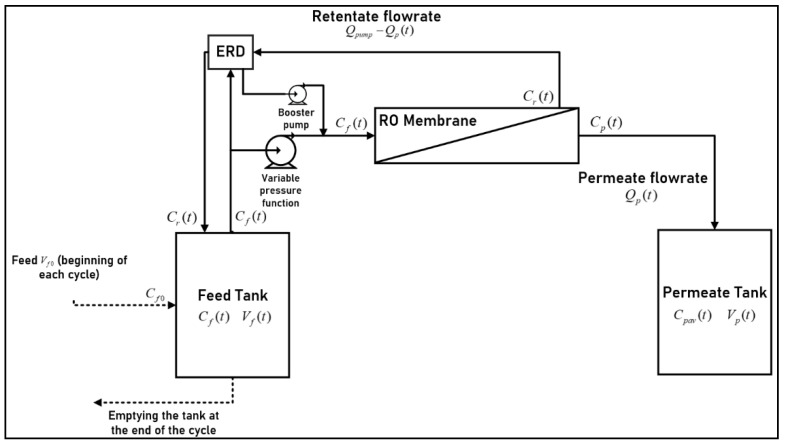
Schematic diagram of the batch RO process.

**Figure 5 membranes-11-00173-f005:**
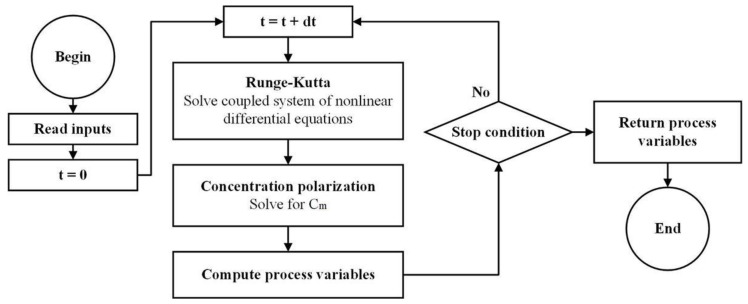
Computation procedure for batch RO including concentration polarization.

**Figure 6 membranes-11-00173-f006:**
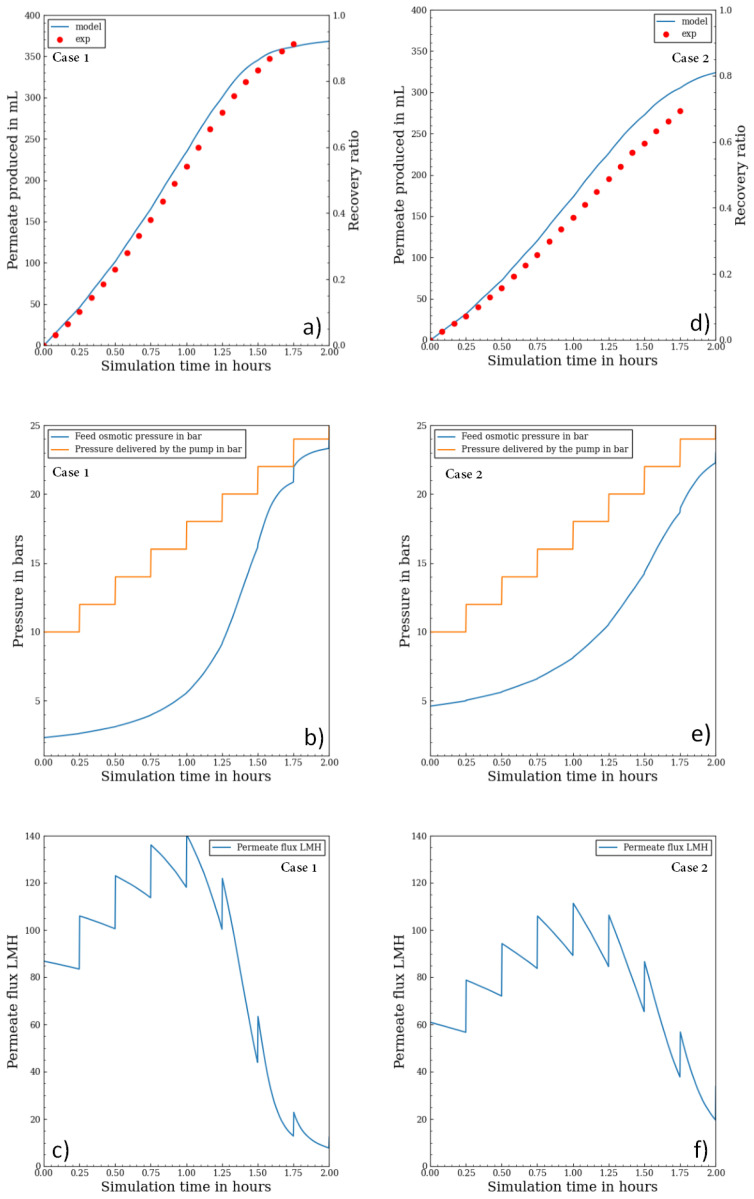
(**a**,**d**) batch RO permeate volume production against time for the model and experiment, (**b**,**e**) batch RO model simulation of osmotic pressure and pump pressure evolution, (**c**,**f**) batch RO model simulation of permeate flux.

**Figure 7 membranes-11-00173-f007:**
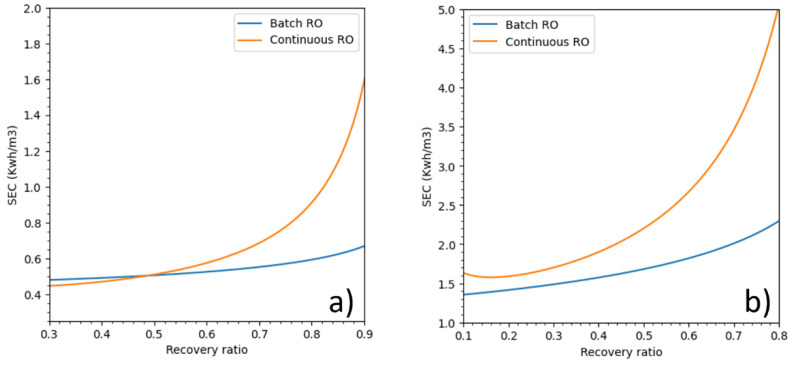
SEC of batch RO and continuous RO vs recovery ratios for (**a**) brackish water and (**b**) seawater.

**Figure 8 membranes-11-00173-f008:**
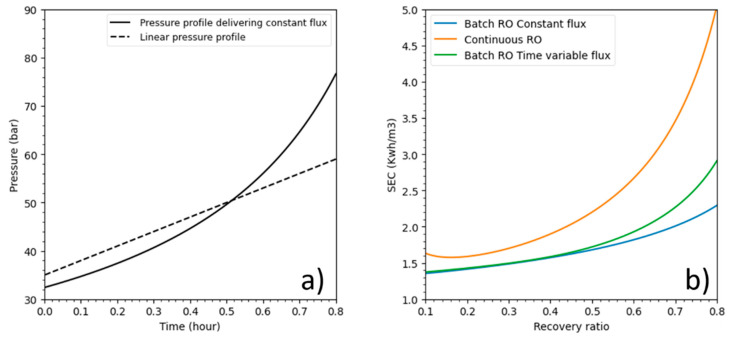
(**a**) Batch RO pressure profiles for seawater (salinity = 35 g/L); (**b**) SEC evolution.

**Figure 9 membranes-11-00173-f009:**
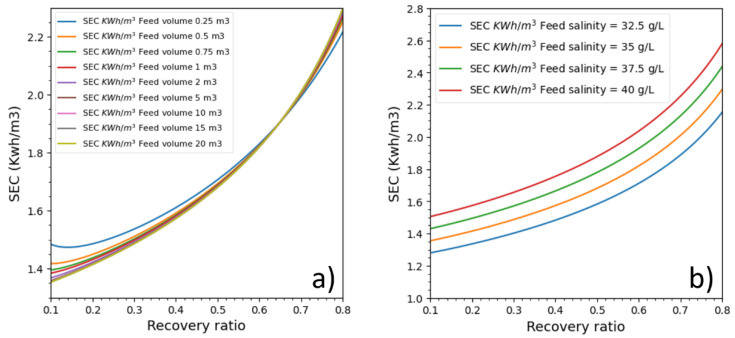
SEC batch RO process for seawater according to (**a**) feed volume and (**b**) feed salinity.

**Figure 10 membranes-11-00173-f010:**
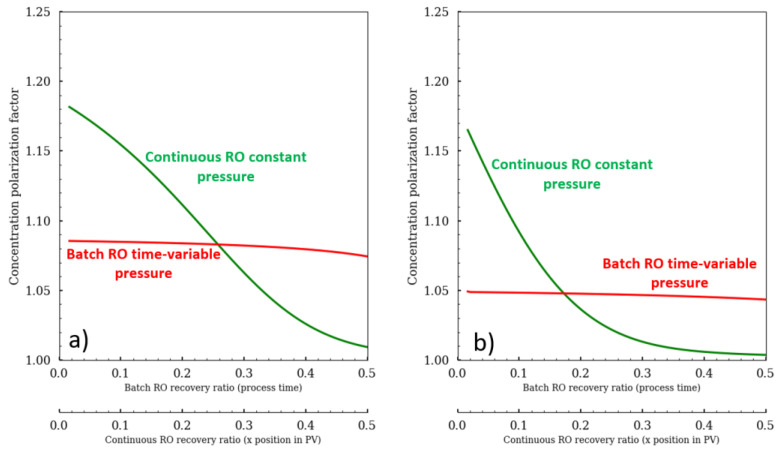
Concentration polarization factor for (**a**) brackish water and (**b**) seawater.

**Table 1 membranes-11-00173-t001:** Parameters used for comparison between the present batch RO model and the model by [14,18].

Parameter	Value (Wei et al.)	Value (This Study)	Units
Intake feed salinity (C_f_)	2–5	2–5	g NaCl/L
Recovery ratios (Y)	29–53	29–53	%
Operating flux	10–20	10–20	LMH
Membrane element area	0.47	0.47	m^2^
Membrane water permeability	4.1	4.1	LMH/bar
Batch RO system volume	2.8	2.8	L
High-pressure pump efficiency	1	1	–
Circulation pump efficiency	1	1	–
Maximum feed pressure	10	10	bar
Circulation pump flow rate	2	2	L/min
Circulation loop pressure drop	0.1	0.1	bar
CPF	Calculated: 1.07–1.14	Calculated: 1.09–1.21	–
k		2.5 × 10^−5^	

**Table 2 membranes-11-00173-t002:** SEC predictions for batch RO, for different permeate fluxes and feed salinities at different recovery ratios, of the present model and of the model from [14,18]. Mean CPF for each case is computed, and the error between the two models is displayed.

Feed Salinity (g NaCl/L)	Permeate Flux (LMH)	Recovery Ratio (%)	Mean CPF Calculated	SEC [14,18] (kWh/m^3^)	SEC Present Study (kWh/m^3^)	Error (%)
2.0	20	28.7	1.21	0.242	0.241	0.41
		38.3	1.21	0.244	0.245	−0.41
		43.1	1.21	0.245	0.247	−0.82
		49.8	1.21	0.249	0.251	−0.80
3.5	10	28.6	1.09	0.251	0.243	3.19
		39.5	1.09	0.256	0.252	1.56
		49.4	1.09	0.263	0.263	0.00
		53.4	1.09	0.267	0.268	−0.37
3.5	15	29.7	1.15	0.264	0.262	0.76
		39.6	1.15	0.268	0.270	−0.75
		49.5	1.15	0.276	0.282	−2.17
		52.3	1.15	0.280	0.285	−1.79
5.0	10	29.7	1.09	0.293	0.291	0.68
		39.6	1.09	0.301	0.304	−1.00
		44.5	1.09	0.307	0.311	−1.28

**Table 3 membranes-11-00173-t003:** Parameters used for energetic comparison between continuous RO and batch RO for seawater and brackish water.

Parameter	Batch and Continuous	Units
Intake feed salinity for seawater	35	g NaCl/L
Intake feed salinity for brackish water	5	g NaCl/L
Membrane element area	37	m^2^
Total elements in system	14	
Elements per pressure vessel	1 for batch and 7 for continuous	
Membrane water permeability	3 for seawater 5 for brackish water	LMH/bar
Operating flux	15 for seawater 25 for brackish water	LMH
Feed tank volume	10 for batch RO	m^3^
High-pressure pump efficiency	0.8	–
ERD efficiency	0.97	–
Pressure drop	0.2 per element	bar
CPF	1	–

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
