# Peer review of "Batch Reverse Osmosis Desalination Modeling under a Time-Dependent Pressure Profile"

_membranes, 2021, doi:10.3390/membranes11030173_

Round 1

Reviewer 1 Report

This paper dicusses the Batch RO system to minimize energy consumption with RO and brackish water. Based on the theoritical apporach, they provide the reduced energy consumption up to 31% for seawater and 19% for brackish water. All the contents are well organized and explained. I don't fee any judegment as well as comments. All the Figure and graphes are good. Therefofore, this paper is well machted for membranes journal without any modification. The published paper will provide good guidline for pressure-driven processs how to control the operation conditions. 

Author Response

We thank reviewer 1 for his positive feedback.

Reviewer 2 Report

To compare the performance and energy consumption of continuous and batch RO processes is very interesting. However, there are many mistakes in mathematical derivation and I can not recommend the publication of this paper.

Equation after equation 12 at line 242: Dimensions on the left and right hand side do not match. Multiplication of Cw(t) made the equation messy.

Equation on line 243 can not be derived from equation 242.

Equation on line 245 can not be derived from equation 243.

Equation on line 248 has a mistake.

Equation 13 can not be derived from equations on line 248 and line 250.

These mistakes may be typo errors. But the results from the equations with so many errors can not be trusted.

Author Response

To compare the performance and energy consumption of continuous and batch RO processes is very interesting. However, there are many mistakes in mathematical derivation and I can not recommend the publication of this paper.

Equation after equation 12 at line 242: Dimensions on the left and right hand side do not match. Multiplication of Cw(t) made the equation messy.

There is a typo error in the submitted version : the equation has been corrected in line 247

We agree that Cw(t) (water concentration in solution) complicates the writing. We simply decided to remove it since it does not affect the calculations assuming Cw= 1 kg/L.

Equation on line 243 can not be derived from equation 242.

It has been deleted and results have been directly presented in the equation of line 250

Equation on line 245 can not be derived from equation 243.

It has been corrected in line 250

Equation on line 248 has a mistake.

It has been corrected in line 255

Equation 13 can not be derived from equations on line 248 and line 250.

It has been corrected in line 262

These mistakes may be typo errors. But the results from the equations with so many errors can not be trusted.

Thank you for pointing these typos errors. We have also revised equations of sections 2.2.3 and 2.2.4.

It should be noticed that these errors were only typos errors and the equations used in the code to perform all simulations were correct.

Reviewer 3 Report

The manuscript presented a process model of batch RO desalination, with consideration of the concentration polarization effect. Batch operation is a promising operating mode that has the potential to enhance the energy efficiency, which deserves more investigation. However, this manuscript needs extensive revision before considering publication. My comments are below:

General comments

  1. The resolution of the figures needs to be improved.
  2. The references are a mess, and references 15 & 20 are missing in the content.  
  3. In section 2.2.2, the equations based on mass balance are from the paper in 1985. The authors should justify why this model in 1985 is selected, compared to recent models. 
  4. Why did the authors choose the fourth order Runge-Kutta method in computation? And the stability and convergence should be discussed.
  5. The units should be consistent throughout the manuscript.

Specific comments:

Line 61, Please add more details on how batch mode reduces concentration polarization.

Line 135-136, Many continuous RO plants can operate at ~80% recovery in groundwater desalination. The conditions need to be more specific when limiting RO recovery to 60%.

Line 173, should not be Vf(t) for permeate tank volume.

Line 180, should be Qr=Qf-Qp.

Line 363, So the first case is treating 6 g/L solution, why higher permeate flux (87) is achieved in case 1 in Figure 1, compared to 60 in second case treating 3 g/L solution? 

Line 366, Why no experimental data presented in Figures 1c and 1f?

Reviewer 4 Report

The paper presents a mathematical model for batch Reverse Osmosis (RO), which allows monitoring all process variables, including specific energy consumption, as a function of time and recovery ratio. Using the considered model, the superior energetic performances of batch RO with respect to continuous RO is shown, especially at a higher recovery ratio. Furthermore, Modeling also proves that batch RO process does not need to work under constant flux conditions in order to show good energetic performance.

The paper is overall interesting and nice to read, and is fully in line with the aims and scope of the journal. Despite its elements of novelty are limited, it addresses a relevant topic and is rich in detail, and therefore, it can be of interest for a wide audience of researchers, technicians and practitioners. Furthermore, the utilization of such model for the analysis of the energetic performance of batch RO compared to continuous one and of the impact of the different factors on energetic responses is very interesting for the potential readers.

The introduction correctly frames the developed research work in the international technical and scientific context, comprehensively describes the addresses topic and the intended objectives and clearly summarizes the contribution of the paper.

In section 2, the experimental set-up is well presented and the adopted model is extensively described. The elements of novelty of the proposed model appear to be quite limited. Nonetheless, such model is described in deep detail and this is a useful aspect for the reader.

The results of the simulations are comprehensively described in Section 3 through both a comparison with experimental results and a comparison with the results of a previous model, which was validated by the experimental batch RO set-up with a pressurized feed tank. The results are meaningful and well represented. However, the quality of Figure 6 needs to be improved, as such figure is difficult to read. Moreover, in the same section a comparison between the energetic performances of continuous and batch RO is proposed, which is the most interesting part of the paper, as the results are relevant from the applicative point of view. Also the analysis of the impact pressure profiles for seawater as well as of feed salinity and feed volume is really interesting in order to understand the constraints of applicability of batch RO in different fields.

The final discussions and conclusions are meaningful and well supported by the provided data and results.

From the formal point of view, page 8 is almost empty and this needs to be avoided. Moreover, the English language is overall good, but a few typos are present (for instance “permeat” instead of “permeate” in page 16 row 498 or the expression “compared with” instead of “compared to” in several occasions), which need to be checked and amended.

Author Response

Comments and Suggestions for Authors

The paper presents a mathematical model for batch Reverse Osmosis (RO), which allows monitoring all process variables, including specific energy consumption, as a function of time and recovery ratio. Using the considered model, the superior energetic performances of batch RO with respect to continuous RO is shown, especially at a higher recovery ratio. Furthermore, Modeling also proves that batch RO process does not need to work under constant flux conditions in order to show good energetic performance.

The paper is overall interesting and nice to read, and is fully in line with the aims and scope of the journal. Despite its elements of novelty are limited, it addresses a relevant topic and is rich in detail, and therefore, it can be of interest for a wide audience of researchers, technicians and practitioners. Furthermore, the utilization of such model for the analysis of the energetic performance of batch RO compared to continuous one and of the impact of the different factors on energetic responses is very interesting for the potential readers.

The introduction correctly frames the developed research work in the international technical and scientific context, comprehensively describes the addresses topic and the intended objectives and clearly summarizes the contribution of the paper.

In section 2, the experimental set-up is well presented and the adopted model is extensively described. The elements of novelty of the proposed model appear to be quite limited. Nonetheless, such model is described in deep detail and this is a useful aspect for the reader.

The results of the simulations are comprehensively described in Section 3 through both a comparison with experimental results and a comparison with the results of a previous model, which was validated by the experimental batch RO set-up with a pressurized feed tank. The results are meaningful and well represented. However, the quality of Figure 6 needs to be improved, as such figure is difficult to read. Moreover, in the same section a comparison between the energetic performances of continuous and batch RO is proposed, which is the most interesting part of the paper, as the results are relevant from the applicative point of view. Also the analysis of the impact pressure profiles for seawater as well as of feed salinity and feed volume is really interesting in order to understand the constraints of applicability of batch RO in different fields.

The final discussions and conclusions are meaningful and well supported by the provided data and results.

From the formal point of view, page 8 is almost empty and this needs to be avoided. Moreover, the English language is overall good, but a few typos are present (for instance “permeat” instead of “permeate” in page 16 row 498 or the expression “compared with” instead of “compared to” in several occasions), which need to be checked and amended.

Thank you for your review and pointing out these improvements.

page 8 is almost empty and this needs to be avoided

The page break in page 8 has been corrected and there is no more a blank page (line 294)

“permeat” instead of “permeate” in page 16 row 498

It has been corrected in line (531)

“compared with” instead of “compared to” in several occasions)

It has been corrected in lines (416, 474)

Round 2

Reviewer 2 Report

The errors in the mathematical derivation were duly corrected.

Reviewer 3 Report

The authors addressed all my concerns.